# Multi-Scale Hydraulic and Petrophysical Characterization of a Heterogeneous Fault Zone in the Gotthard Massif's Crystalline Basement

Tom Schaber<sup>1</sup>, Mohammedreza Jalali<sup>1</sup>, Alberto Ceccato<sup>2</sup>, Alba Simona Zappone<sup>3</sup>, Giacomo Pozzi<sup>4</sup>, Valentin Gischig<sup>5</sup>, Marian Hertrich<sup>6</sup>, Men-Andrin Meier<sup>6</sup>, Timo Seemann<sup>1</sup>, Hannes Claes<sup>7</sup>, Yves Guglielmi<sup>8</sup>, Domenico Giardini<sup>9</sup>, Stefan Wiemer<sup>5</sup>, Massimo Cocco<sup>4</sup>, Florian Amann<sup>1,10</sup>

Correspondence to: Tom Schaber (schaber@lih.rwth-aachen.de)

## Abstract.

Accurately characterizing fault zones in crystalline basement rocks is essential for understanding fluid migration in the Earth's crust and how this influences fault stability and seismicity. While it is known that fault zones exhibit strong heterogeneity in structure and hydraulic properties, quantifying these variations across scales remains a challenge. The study presented investigates a deeply buried fault zone intersected by two inclined boreholes within a high overburden underground research laboratory (URL). As part of the FEAR (Fault Activation and Earthquake Rupture) project, this work provides key hydraulic and structural constraints needed to select and prepare experimental injection sites. These findings pose a necessary foundation for developing controlled fluid injection experiments and emphasize the importance of understanding scale-related effects during multi-scale observations. Through a combination of field-scale hydraulic testing, geophysical logging, and petrophysical analyses of core samples, we evaluate permeability, porosity, wave velocities, and fracture characteristics across multiple structural facies and on varying scales. The study finds that permeability varies over several orders of magnitude, largely controlled by the presence and connectivity of open fractures. Comparisons between lab and field data reveal pronounced scale effects, with lab tests underestimating the in-situ permeability due to the exclusion of large fractures and structural discontinuities. The fault zone shows a combination of localized and distributed flow behaviours, with no evidence of a continuous low-permeability fault core.

<sup>&</sup>lt;sup>1</sup>RWTH Aachen University, Engineering Geology and Hydrogeology, Aachen, Germany

<sup>&</sup>lt;sup>2</sup>Department of Earth Sciences, Structural Geology and Tectonics Group, Geological Institute, ETH Zurich, Switzerland

<sup>&</sup>lt;sup>3</sup>Institute of Geology, ETH Zurich, Zurich, Switzerland

<sup>&</sup>lt;sup>4</sup>Istituto Nazionale di Geofisica e Vulcanologia, Roma, Italy

<sup>&</sup>lt;sup>5</sup>Swiss Seismological Service, ETHZ, Zurich, Switzerland

<sup>&</sup>lt;sup>6</sup>Institute of Geophysics, ETH Zurich, Zurich, Switzerland

<sup>&</sup>lt;sup>7</sup>Department of Earth and Environmental Sciences, KU Leuven, Celestijnenlaan 200E, 3001, Heverlee, Belgium

<sup>&</sup>lt;sup>8</sup>Lawrence Berkeley National Laboratory, 1 Cyclotron Road, Berkeley CA, 94720, USA

<sup>&</sup>lt;sup>9</sup>Department of Earth Sciences, ETH Zürich, Sonneggstrasse 5, 8092 Zürich, Switzerland

<sup>&</sup>lt;sup>10</sup>Fraunhofer Research Institution for Energy Infrastructures and Geotechnologies IEG, Aachen, Germany

# 1 Introduction

55

Earthquakes are one of the most fatal appearances among natural disasters. Worldwide, out of the ten natural disasters between 1995 and 2022 causing the most deaths, seven were earthquakes (Tin et al., 2024). Besides the immediate deaths and casualties, they often trigger delayed secondary hazards such as landslides, soil and debris flows, tsunamis, or they pose a risk by disrupting critical infrastructure such as healthcare and food supply systems, potentially leading to localized societal breakdowns, which claim further lives (Tin et al., 2024). Although a certain understanding of what causes these natural tremors exists, past and state-of-the-art scientific methods and knowledge do not enable a reliable earthquake forecast or prediction (Geller, 1997; Mizrahi et al., 2024; Navarro-Rodríguez et al., 2025). Since the nucleation depth of most earthquakes lies several kilometres below the surface, their wave propagation is scattered and attenuated before reaching any seismometer, thus losing parts of the information the waves initially carried (Achtziger-Zupančič et al., 2024). Therefore, it poses a fundamental challenge to track and record these very early processes during nucleation and early rupture. In addition, the fault planes' slip, deformation, stability, permeability, and reactivational behaviour are further aspects that contribute to this complex system.

Past research approached this topic on numerous scales. From laboratory studies on the centimetre and decimetre scale (Cappa et al., 2018, 2019; Ji et al., 2021, 2022; Passelègue et al., 2018; Scuderi et al., 2017; Volpe et al., 2023; Wang et al., 2020; Ye and Ghassemi, 2018), to field projects reaching dimensions in the decametre range (Amann et al., 2018; De Barros et al., 2018; Evans et al., 2004; Guglielmi et al., 2015; Kakurina et al., 2019, 2020). Most of these field projects reach depths of several hundred meters below the surface, therefore not quite attaining realistic in-situ conditions that would be found along a deeply buried fault plane under seismogenic conditions. The Bedretto Tunnel in Ticino, Switzerland, hosts an Underground Laboratory which offers access to deep crystalline fault zones with an overburden of 1-1.5 km (Ma et al., 2022). This enabled the realisation of the FEAR project, aiming at deliberately inducing controlled earthquakes via hydraulic stimulation. Injecting in and around a fault core, for a sufficient amount of time with the correct pressures, flow rates, and possibly some preconditioning, should trigger a seismic event, causing the fault plane to slip. A dense multidisciplinary monitoring network in the vicinity of the fault plane traces seismicity, strain, pore pressures, and tilt, among other parameters. This enables high-resolution, close-up observation and monitoring of the stimulated rock volume before, during, and after tremors occur.

To develop an appropriate injection scheme and protocol for the stimulation experiments, the fault zone must be hydraulically and structurally characterized beforehand. Therefore, a double-straddle packer system was deployed in two boreholes intersecting the fault zone. This allowed a hydraulic characterization and provided a collection of in-situ parameters such as permeability, pore pressure, or injectivity. In addition to the field campaign, specimens from one borehole were retrieved and tested in the laboratory using a suite of petrophysical techniques. The comparison of field and laboratory data illustrates a scale effect, highlighting the importance of being aware of the measurement dimension involved. This characterisation marked the foundation for all injection experiments in this rock volume. In addition to the purely scientific research questions regarding

the seismological and thermo-hydraulic-mechanical aspects of the FEAR project, its insights might be useful for other areas, such as the deep geothermal industry that has to deal with induced seismicity.

# 2 Geological setting

Located in the Swiss Central Alps, the Gotthard massif stretches about 80 km in WSW-ENE and 10 km in NNW-SSE direction (Rast et al., 2022). Throughout the opening of the Paleo-Tethys ocean in Triassic times, the Gotthard massif underwent extensional tectonics (Guillot and Ménot, 2009). During the Early to Mid-Cenozoic, the massif was affected by the Alpine orogeny, reaching peak metamorphic conditions of over 550 °C and 0.9 GPa (Ceccato et al., 2024). The convergence of the European and Adriatic plates facilitated the rapid exhumation of the crystalline units of the Gotthard-Aar massif between approximately 22 and 17 million years ago (Challandes et al., 2008; Goncalves et al., 2012; Herwegh et al., 2017; Oliot et al., 2010; Rolland et al., 2008, 2009). During the Mid-Miocene, the Gotthard massif was predominantly influenced by regional strike-slip tectonic processes (Campani et al., 2010; Herwegh et al., 2017; Rolland et al., 2009). In contrast, during the Late Miocene, it underwent shallow-level brittle deformation, resulting in the reactivation and development of gouge-bearing brittle fault systems (Kralik et al., 1992; Pleuger et al., 2012). The Gotthard massif hosts several magmatic bodies (Fig. 1). One of these intrusions, the Rotondo granite, is a post-Variscan igneous rock body (Berger et al., 2017) that hosts the largest part of the Bedretto Tunnel and dates back to the Early Permian age, 295 Ma ago (Rast et al., 2022). Upon entering the tunnel from its south portal in Ronco, the younger pre- and early-Variscan Tremola and Prato series will be encountered first. Both the Tremola and Prato series (Pontino zone and Val Nalps, in Fig. 1) are primarily characterized by metasedimentary and some metaigneous lithologies (Hafner, 1958; Keller et al., 1987; Steiger, 1962), with the Prato series having a more complex internal structure due to steep isoclinal folds (Rast et al., 2022). After tunnel meter 1138 (abbreviated TM1138; seen from the Ronco 85 entrance), the next occurring lithology is the Rotondo granite, which extends until the tunnel's northwestern end at the Furka Base Tunnel, located at TM 5218 (Keller and Schneider, 1982). The Rotondo granite is mostly fine-grained and equigranular (RG1). At the same time, it be biotite-rich and porphyroclastic (RG2) (Schneider, 1985), which is mainly encountered in the middle tunnel section, specifically between TM2805 and TM3437 (Achtziger-Zupančič et al., 2024). Regarding their mineral composition, RG1 and RG2 do not differ, whereas the proportion of their phases varies (Rast et al., 2022). It was during the rapid exhumation in the Early Miocene that the Rotondo granite received its NE-SW and ENE-WSW-trending ductile shear zones, which are present to this day. These shear zones developed on a pre-existing occurrence of brittle faults and shear fractures (Ceccato et al., 2024).

Figure 1: (a) Overview of a section of the Gotthard massif, including the Bedretto Tunnel, and the Rotondo granite among other intrusion bodies. (b) Cross-section of the Bedretto tunnel, the Pizzo Rotondo, and all significant features along the tunnel axis. Modified after Achtziger-Zupančič et al., 2024.

#### 100 **2.1 Test volume**

The two boreholes that this study will predominantly focus on are BFE\_A\_05 and BFE\_A\_06 (see Fig. 2). For the sake of clarity, the boreholes will be referred to as boreholes BO5 and BO6 for the remainder of the paper. Their borehole mouths are directly juxtaposed at TM2363. BO5 and BO6 have azimuths and dips of  $31.5^{\circ}/29.5^{\circ}$  and  $24.7^{\circ}/10.3^{\circ}$ , both dipping downwards. Their respective lengths are 216 and 55 m, both with a diameter of 96 mm. BO5 and BO6 intersect the target fault structure (MC fault) at approximately 37-45 m and 24-29 m from the borehole mouths, respectively. Figure 2 highlights a 14 and 10 m section of the individual boreholes capturing features in and around the MC fault core. BO5 and BO6 were both hydraulically tested using a double-straddle packer system (see intervals in Fig. 2). For lab testing, core samples were retrieved from BO5 only. Figure 2 shows all discontinuities identified from acoustic and optical televiewer data and manual core logging, classified as open fractures with visible apertures or undifferentiated fractures without a discernible aperture. This classification is essential for interpreting the hydraulic test results.

Stress measurements performed during a mini-frac campaign (Bröker and Ma, 2022) provide essential information about the prevailing stress field, including its magnitudes and orientation. This is required to determine the correct effective stresses for lab testing. The mini-fracs were conducted around the Bedretto Lab at TM2000 (TM1750-TM2250). Hence, due to its proximity to the FEAR experiments at approximately TM2400, we assume that the inferred stress field also applies to our test volume (Achtziger-Zupančič et al., 2024). The derived stresses indicate a transitional regime between normal and strike-slip faulting with:  $\sigma_{v} \approx 26.5$  MPa,  $\sigma_{Hmax} = 20.4 - 27.9$  MPa, and  $\sigma_{hmin} = 11.2 - 16.4$  MPa. Orientation of  $\sigma_{Hmax}$  varies between N100°E and N120°E (Fig. 2).

- The fault zone exhibits heterogeneous physical and geometrical characteristics. Therefore, several core facies that occur throughout all intersecting boreholes in the volume were identified. These facies were classified based on their geological characteristics and compared to observations along the tunnel wall, as the fault structure also intersects the Bedretto tunnel itself. The facies allow the breakdown of the geometry and occurrence of the fault zone at depth. They are defined as follows (Achtziger-Zupančič et al., 2024):
  - Facies 0 (F0): intact rock with rare (

Figure 2: OPTV and ATV of boreholes 5 and 6. Picked structures, hydraulic testing intervals, sampling locations, and facies occurrences are highlighted. Stereonets with discontinuities and stress state are indicated.

Facies occurrences along the boreholes are depicted in Fig. 2. Note that F2 is missing in BO6, and how the overall spatial extent of the fault structure thickness varies between the boreholes. This is mainly due to the different angles of intersection and obliqueness between BO5 and BO6 and the average orientation of the fault structure. Three main sets of lineaments were identified across the mountain range via remote sensing (Achtziger-Zupančič et al., 2024). Set (1) consists of NE-SW to ENE-WSW striking lineaments with a lateral extent of several hundred meters. Set (2) features N-S trending lineaments with limited lateral persistence, and set (3) comprises WNW-ESE to NW-SE trending lineaments, also with limited lateral extent.


All relevant discontinuities were characterized and described by Ceccato et al. (2024). They found four distinct deformation structures, which were chronologically classified as D1 (oldest) to D4 (youngest). These discontinuities are present throughout the tunnel and within the investigated rock volume of this contribution. Each of the four types of deformation structures can be assigned to one of the previously mentioned lineament sets.




The Rotondo granite underwent a brittle-ductile-brittle development, spanning from the pre-Alpine orogeny to being overprinted during peak and post-convergent tectonics (Ceccato et al., 2024). The target fault system, known as the MC fault, can be classified as a type D4 structure, with an expected lateral extent exceeding 100 m (Achtziger-Zupančič et al., 2024). Type D4 discontinuities are part of lineament set (1), which, on outcrop scale, are organized in clusters that comprise discrete and discontinuous shear and fracture planes with less than 30 m length (Achtziger-Zupančič et al., 2024). The stereonets in Fig. 2 primarily display lineament set (1) with type D4 discontinuities. Mylonitic shear zones can be encountered at 45.2 m in BO5 and at 28.4 m and 29.2 m in BO6. Figure 3 presents a plan view of the test volume, highlighting all key structures. The sampling distribution in BO5 was designed to encompass all different facies. If feasible, the samples for lab testing originated from the locations where the hydraulic testing intervals were situated, allowing for a direct comparison at lab (cm) and in-situ (dm-m) scales.

Figure 3: Overview of all FEAR-related structures in plan view around TM2400. Modified after Achtziger-Zupančič et al., 2024.

# 170 3 Methodology




# 3.1 Laboratory work

#### 3.1.1 Permeability

Determination of laboratory-scale permeability is a key element in petrophysics. Retrieved samples of different facies along BO5 were overcored and cut into cylindrical plugs with a diameter of 38 mm, while plug lengths varied from 18 to 33 mm. A total of eleven samples were measured in the flow cell. The setup was powered by two separate piston pumps: one served as the confining pump and the other as the pore pressure pump, utilizing water as the working fluid. This enabled simultaneous and independent control of the two pumps. As the thin capillaries of the setup are highly sensitive to ambient temperature changes, both room temperature, confining, and pore pressure fluids needed to be kept at a constant value of 25 °C. In addition to the closed flow cell system, a burette was attached, allowing for the controlled release of water for steady-state measurements (see Fig. 4 (a)). As previously mentioned, the effective stresses chosen for this laboratory campaign are based on those proposed by Bröker and Ma (2022). Effective isotropic stresses of 15, 25, and 32 MPa were applied throughout all tests. The different effective stresses enabled to capture hydraulic behavior of a plug, regardless of its original orientation in its in-situ state. A volumetric calibration and leak test were conducted before the start of measurements. As the samples exhibited strong macroscopical heterogeneities, the plugs were expected to have a wide range of permeability. Undisturbed granite plugs consisting of a tight, crystalline matrix are expected to be of low permeability, as many authors have observed in the past


(Bischoff et al., 2024; Brace et al., 1968; David et al., 2018, 2020; Mitchell and Faulkner, 2008; Osten et al., 2024; Selvadurai et al., 2005; Wenning et al., 2018). Therefore, this study employed two different measurement techniques, allowing for the coverage of a broader range of permeability.

For low-permeable samples, the pulse decay technique was introduced (Brace et al., 1968). This method covers a permeability range of approximately 10<sup>-16</sup>-10<sup>-21</sup> m<sup>2</sup>. Besides the actual permeability, the fluid type, the chosen pore pressures, and the plug length are key factors in controlling the measurement duration. The pulse decay method is a transient measurement technique that does not rely on a constant flow rate or pressure gradient. Analysing the pressure decay curves of both reservoirs and applying regression to the curves enables the calculation of the sample's permeability (see Fig. 4 (b)).

For samples with higher permeability, the steady-state approach was employed. A steady pore pressure was applied to the bottom reservoir and maintained until steady-state equilibrium conditions were achieved. This includes a steady flux of pore fluid leaving the sample while the pressure gradient between the system and the atmosphere remains constant. By applying linear regression to the steady-state part of a graph depicting flow volume against time (see Fig. 4 (c)) and using Darcy's law, the permeability can be calculated.

Figure 4: (a) Laboratory flow cell setup for cylindrical plugs. (b) Principle of pulse decay technique. (c) Steady-state flow experiment requiring constant flow and pressure as underlying conditions.

# 205 **3.1.2 Porosity**




Water immersion porosimetry (WIP) was employed to determine porosity. This approach was specifically used to measure core segments and plugs, aiming to achieve a scale effect. Applying WIP requires determining the dry mass, saturated mass, and the mass of the saturated sample immersed in deionized water. Using these parameters, one can compute the water-saturated bulk density and the anhydrous grain density, enabling the calculation of the effective porosity (Kuila et al., 2014). The significant advantage of this method lies in its simplicity and the possibility of measuring specimens with nearly unrestricted sizes. As a second porosity-determination technique, this study employed He-pycnometry. This is based on Boyle's Law for ideal gases and utilizes the principle of cyclic gas expansion. A valve connects two reservoirs of known volume. In one of the reservoirs, the specimen is placed. The rock skeletal volume can be determined by repeatedly expanding gas from the reference reservoir into the sample reservoir.

#### 215 **3.1.3 P-wave velocity**

P-wave velocity was measured using a multi-sensor core logger (MSCL) on various core segments. Subsequently, the retrieved plugs from the cores were measured with an acoustic emission (AE) system. Both devices measured the samples in dry, non-confined conditions. The sampling line along the longitudinal axis of a core was chosen to intersect as many structural features as possible at the most orthogonal angle feasible. Mechanical waves propagating subparallel to a media boundary will not reflect or refract as noticeably as they would when hitting an orthogonally oriented boundary. The MSCL used a centre frequency of 230 kHz and a spatial sampling rate of 1 cm<sup>-1</sup>. After the plugs were overcored from the core segments, they were measured with an AE system, allowing for much smaller specimen sizes. Ten pulses with 300 V were deployed for each plug. All ten waveforms were stacked and averaged, allowing for the picking of the signal onset at the receiving end.

# 3.2 Field work

# 225 3.2.1 Geophysical logging

BO5 and BO6 were logged using an optical (OPTV) and acoustic televiewer (ATV). Combining both allows for identifying open fractures with visible apertures and fractures with infillings. This was necessary to decide where to place the hydraulic testing interval. As both boreholes are negatively inclined, they were filled with water, allowing the use of an ATV. In addition to the log images, a full wave sonic (FWS) survey was carried out, providing the compressional (Vp) and shear wave (Vs) velocities (in BO5, only Vp was measured). This gave an in-situ understanding of the mechanical wave properties, in addition to the laboratory data. The empirical relationship of Gardner et al. (1974) was used to infer density from Vp, with  $\alpha = 0.37$  and  $\beta = 0.23$  in Gardner's equation (2). Subsequently, established reference laboratory densities (David et al., 2020; Osten et al., 2024) were used to calculate porosity along the boreholes.

$$p = \alpha V_p^{\ \beta} \tag{2}$$





# 3.2.2 In-situ hydraulics

Hydraulic testing was conducted in both boreholes BO5 and BO6. A total of eleven intervals were tested, spanning across the fault zone and encompassing all different lithologies and discontinuities (see Fig. 2 for interval placement). A double-straddle packer system with an interval length of 0.73 m and a packer length of 1 m was used. The testing sequence generally obeyed the following order: first, a phase of pressure equilibration that allowed the interval to saturate and build up its in-situ prevailing pore pressure. Second, pulse injection tests provided an initial indication of the packer integrity, system compressibility, and interval transmissivity. If the decay of the pulse indicates a sufficiently high permeability, the next step would be to conduct a constant rate test. Therefore, water was injected at a constant flow rate (10 to 0.1 L min<sup>-1</sup>) into the interval over approximately 20 minutes, after which a sudden stop of the pump induced a shut-in. Usually, a series of pulse and rate injection tests was conducted for a single interval. Around the fault core, where fracture intensity was high, interval placements often only permitted overlapping packers and fractures. Figures 5 (a) and (b) depict such tests.

The hydraulic tests were evaluated using the n-dimensional Statistical Inverse Graphical Hydraulic Test Simulator, nSIGHTS

(Sandia National Laboratories (SNL), 2012). Additional code was used for the type curve-based approach (Bredehoeft and Papadopulos, 1980), as well as for constant rate tests, and shut-in (Cooper and Jacob, 1953; Horner, 1951). The evaluation approaches are illustrated in Fig. 5 (c), (d), and (e). All borehole hydraulic tests provide results in either transmissivity (m s<sup>-2</sup>) or hydraulic conductivity (m s<sup>-1</sup>). All hydraulic data must be converted into a single unit to be able to compare laboratory and field results. Therefore, field values were first converted from transmissivity (m<sup>2</sup> s<sup>-1</sup>) to hydraulic conductivity (m s<sup>-1</sup>), and then to permeability (m<sup>2</sup>). The underlying assumptions for these calculations, in addition to the known interval thickness, were a water density of 1000 kg m<sup>-3</sup> (National Institute of Standards and Technology, 2025), an average gravitational acceleration of 9.81 m s<sup>-2</sup>, and a dynamic viscosity of water of 1 mPa s<sup>-1</sup>, all at 20 °C (Berstad et al., 1988).

Pore pressures were determined at the end of equilibration phases before testing. If the interval did not fully reach its ambient static pore pressure, it was computed using the extrapolated y-intercept of the curve regression of the Horner plot. Additionally, injectivities were calculated using data from another testing campaign with a larger 2.4 m interval, which included step-rate tests that enabled the creation of P-Q plots. Injectivity refers to the volume of fluid that can permeate the rock under a given injection pressure ( $m^3$  s<sup>-1</sup> MPa<sup>-1</sup>, (Martínez et al., 2021)). Some expected structures that link parts of the two boreholes, potentially both structurally and hydraulically, were also tested. Therefore, pressure and flow rates in the injecting and monitoring boreholes must be surveyed. Under steady-state conditions, and with knowledge of the radial distance from the pumping well r, the borehole radius  $r_w$ , the hydraulic head gradient  $\Delta\Phi$ , and the discharge rate of the pumping well  $p_w$ , one can determine the transmissivity, thereby estimating the permeability using Thiem's equation (Thiem, 1906):

$$T = \frac{p_W}{2\pi\Delta\Phi} \ln(\frac{r}{r_W}) \tag{1}$$

Figure 5: (a) Pressure and flow over time as a pulse injection is performed. (b) Depiction of pressure and flow throughout a constant rate test, with subsequent shut-in. (c) Evaluation approach with type curves for pulse or slug tests, after Bredehoeft and Papadopulos. (d) Evaluation approach for constant rate testing with decade regression after Cooper-Jacob. (e) Shut-in recovery evaluation after Horner.

Flow dimensions were computed using the Generalized Radial Flow Model (Barker, 1988). It describes how injection pressure changes during a constant-rate hydraulic test and reveals the effective geometry of flow in fractured rock. Instead of assuming perfectly radial spreading, GRF introduces the flow dimension, which can vary between 1 (linear, channelized flow) and 3 (spherical flow). The slope of pressure versus time in log–log space can be used to infer the flow dimension. In order to avoid wellbore storage or boundary effects in the very early or late pumping phases, the correct time window must be chosen for evaluation.

# 4 Results

#### 4.1 Field scale

#### 4.1.1 BO5

290 Between meters 34 and 48, BO5 encompasses all known facies, as defined by Achtziger-Zupančič et al. (2024). Figure 6 shows all logs from BO5. The caliper displays significant disturbances at around 38 m, overlapping with the position of facies three. No further caliper anomalies are visible along the rest of the depicted borehole. The mechanical waves in the FWS log are partially interrupted, resulting in data gaps. This may have been caused by poor tool eccentricity or intense fracturing with significant apertures. Vs is relatively constant throughout the entire log, with data gaps at depths of approximately 39-40 m 295 and 41-42 m, and an average value of approximately 2.79 km s<sup>-1</sup>. Vp is reaching peak velocities of over 7.50 km s<sup>-1</sup> and lowest values of ca. 3.30 km s<sup>-1</sup>, with an average of 5.05 km s<sup>-1</sup>. Single data point spikes in velocity can be ambiguous, as they may indicate xenoliths or larger crystals. The data gaps in Vp appear to be concentrated around the fault core. Reference data from literature on similar lithologies indicate average values for Vs and Vp of 3.40-3.50 km s<sup>-1</sup> and 5.30-5.40 km s<sup>-1</sup>, respectively (David et al., 2020; Wenning et al., 2018). The fault zone's fracture intensity reduces compressive and shear wave velocities. The permeability in BO5, as determined by hydraulic testing across the fault zone, has mean values ranging from  $1.6 \times 10^{-11}$ 300 to  $5.6 \times 10^{-16}$  m<sup>2</sup>. Permeabilities shown as boxplots in Fig. 6. Intervals are labeled 1 to 7, from top (34 m) to bottom (48 m). The highest permeability occurs in interval 2, and the lowest in interval 3.

The low-permeability interval only allowed for pulse testing, as the pump could not sustain a sufficiently small flow for constant rate testing. Despite interval 3 containing visible fractures, it is less permeable than any other interval in BO5. This can be explained by the interval containing only partially open fractures, and possibly some infillings and mineralization. Additionally, the packer placement may not have been entirely accurate, resulting in a wide aperture fracture at 39.80-40.10 m, which might have fallen just outside the specified interval. These assumptions, combined with the fact that all other intervals contain at least one continuous and non-infilled fracture with clear aperture, result in a visibly reduced permeability. Moving away from the fault core beyond 40 meters, the fifth interval has an average permeability of  $4.5 \times 10^{-14}$  m<sup>2</sup>, while the sixth interval is slightly higher due. The last interval between 45 and 46 meters depicts an interval that barely intersects with a wide-open aperture fracture. The overlap is not large enough, or the fracture does not allow water to pass through; therefore, its permeability averages  $1.7 \times 10^{-15}$  m<sup>2</sup>. The amount of open fractures in the linear fracture intensity  $P_{10}$  is highest around the fault core, while it decays abruptly towards the top and slowly over several meters towards the bottom. As porosity was inferred from density, or empirically, from Vp, the same data gaps from Vp can be observed in the porosity subplot. Therefore, the region around the fault core was not captured and cannot be interpreted. The average porosity is approximately 3.28 %, with minimum and maximum values of 0.51 % and 14.74 %, respectively.

Figure 6: Different logs of borehole BO5 including: facies, caliper, Vp and Vs, permeability boxplots, linear fracture intensity P<sub>10</sub>, porosity, static pore pressure, and injectivities. White hollow circles indicate permeability outliers.

Facies four with vuggy pores does not show the expected higher porosity. Pore pressures are highest in the fault core, with a decrease of more than 2.5 MPa in the neighbouring third and fourth intervals. The pore pressure settles between 1 and 2 MPa in intervals five, six, and seven. The maximum injectivity can be found in the fault core with around  $1.1 \times 10^{-4}$  m<sup>3</sup> s<sup>-1</sup> MPa. The borehole exhibits similar high injectivities in facies zero if open fractures with large enough apertures are present.





# 4.1.2 BO6

Unique features of BO6 are that it does not contain facies two, and facies four appears both above and beneath the fault core. All logs from BO6 are displayed in Fig. 7. The apparent overall fault zone thickness seems smaller than in BO5. Besides actual differences in the structural architecture, the less oblique angle between the borehole plunge and the average orientation of the fault zone plane can cause this observation. The caliper shows two main areas of disturbance. One area is approximately 24.5 to 25.0 m, while the other is located close to the fault core, between 28 and 29 m. Vp is partly discontinuous, and the data do not indicate extensive areas of higher or lower velocities. Its minimum and maximum values are 4.16 km s<sup>-1</sup> and 6.66 km s<sup>-1</sup>, respectively, with a mean Vp of 5.22 km s<sup>-1</sup>. The first interval in BO6 has a mean permeability of  $3.4 \times 10^{-14}$  m<sup>2</sup>. The second interval belonging to facies three has a mean permeability of  $1.1 \times 10^{-13}$  m<sup>2</sup>, and the third one situated in the fault core averages  $1.2 \times 10^{-11}$  m<sup>2</sup>, but also shows the most extensive data spread. The fourth interval has a permeability of  $4.9 \times 10^{-14}$  m<sup>2</sup>. The fracture intensity shows that, like in BO5, the fault core exhibits the highest fracturing. In contrast, the distribution of open fractures appears to be more diffuse compared to BO5. Overall, fracture intensity is slightly increased in BO6 compared to BO5. Mean porosity is approximately 2.0 %, and does not yield any significant trends, just as Vp, and does not reflect the fracture behaviour of this borehole. This may be caused by varying apertures or fracture infillings, which distinguish the two boreholes. Pore pressure is lower than in BO5, from 2 MPa in the fault core to 1.37 MPa in the intensely fractured zone at 25 m. Injectivities yield similar results to BO5, except that the highest value is not achieved in the fault core, but mainly in facies four. Facies occurrences around the fault core in BO6 have significantly smaller thicknesses, making it difficult to isolate single facies properly (see interval placement in Fig. 2). Therefore, individual facies hydraulics in BO6 should be cautiously interpreted, as they might have resulted from intervals overlapping multiple facies.



Two values for the connected structures between the two boreholes, BO5 and BO6, were determined. As this data originates from another testing campaign, other equipment was used during experiments. The interval size was 2.4 m. First, an injection at 43.2-45.6 m in BO5 was conducted while measuring pressure and outflow in BO6. This resulted in a permeability of  $1.6 \times 10^{-15}$  m<sup>2</sup>. A second test included an injection at 26.0-29.7 m (3.7 m interval) in BO6, while monitoring pressure and outflow in a packed-off interval in BO5 at 36.6-39.0 m, encompassing the fault core in both boreholes. This yielded  $3.9 \times 10^{-15}$  m<sup>2</sup>.

Figure 7: Different logs of borehole BO6 including: facies, caliper, Vp, permeability boxplots, linear fracture intensity  $P_{10}$ , porosity, static pore pressure, and injectivities. The white hollow circle indicates a permeability outlier.


# 4.2 Laboratory scale

# 4.2.1 Permeability

Since most plugs were overcored at an oblique angle to the core longitudinal axis, the plugs approximately span 4 cm across a core. Hence, most plugs are named after a 4 cm section. Only samples from BO5 were retrieved. Figure 8 (a) shows permeability of facies one. Samples 35.04-35.08 m and 41.52-41.56 m are undisturbed, while 41.31-41.35 m has a closed biotite-filled shear fracture. Under consideration of all effective stress stages, the permeability in facies one varies from low  $10^{-18}$  m<sup>2</sup> down to  $2.0 \times 10^{-19}$  m<sup>2</sup>, with the specimens varying the strongest at 32 MPa effective stress. Figure 8 (b) depicts facies two with the undisturbed sample 39.19-39.23 m and 40.02-40.06 m, which contains some thin chlorite and quartz-dominated fault gouge. These specimens vary in permeability from  $9.0 \times 10^{-18}$  m<sup>2</sup> to  $2.5 \times 10^{-18}$  m<sup>2</sup>. Facies three is shown in Figure 8 (c) representing the fault core. Samples 37.50-37.54 m and 37.53-37.57 m contain closed shear fractures and thin

Figure 8: Influence of effective stress on permeability of 11 samples. (a) Facies 1, (b) Facies 2, (c) Facies 3, (d) Facies 4. Each facies is accompanied by images of the core segments from which the plugs originate.

sericite veins. 38.49-38.53 m is intact and undisturbed, and 38.58-38.62 has a partial open fracture with biotite and sericite infilling. The maximum and minimum permeability range from  $3.5 \times 10^{-17}$  m<sup>2</sup> to  $4.0 \times 10^{-18}$  m<sup>2</sup>. The fourth facies (Fig. 8 (d))





has macroscopically visible cavities, likely due to hydrothermal alteration. Sample 43.35-43.39 m contains an open shear fracture, while 43.40-43.44 m has partially open fractures with biotite and quartz infillings. The specimen with the open shear fracture in combination with its vuggy structure attains a permeability of  $2.0 \times 10^{-13}$  m<sup>2</sup>, which is the highest permeability out of the total 11 samples measured. The lowest determined permeability is  $5.5 \times 10^{-17}$  m<sup>2</sup> of facies four.

Comparing these values shows that effective stresses more intensely impact tighter facies than samples with an already high baseline permeability. This is because grain boundaries or thin fractures in altered samples can quickly be compacted. In contrast, open shear fractures or vuggy cavities are often too voluminous to be strongly affected by increasing effective stress. Similar findings were reported in Evans et al. (1997).

# 4.2.2 Porosity and P-wave velocity

The lowest and highest porosity of all core segments using WIP were approximately 0.8 % and 5.0 %, respectively, at 41.20-41.38 and 43.30-43.50 m. The minimum and maximum WIP porosity for plugs were 0.7 % and 6.3 % at positions 35.19-35.23 m and 43.35-43.39 m. Figure A1 (appendix) compares the porosities of entire core segments and plugs, indicating that in lower disturbance facies, plugs display slightly higher porosity than the core segments they originate from. In stronger disturbed facies, this trend is not visible. WIP underestimates absolute porosity compared to He-pycnometry, but can still accurately capture relative porosities in a test series. The lowest and highest average Vp of cores were determined to be 2.49 km s<sup>-1</sup> and 4.08 km s<sup>-1</sup> at 38.55-38.75 m and 43.40-43.50 m. No relationship was found concerning facies type and Vp on different scales, as discontinuities were tested at different orientations on the core and plug scale (see Fig. A2).

# 5 Discussion

# 390 5.1 Scale effects

Observations at different scales are required to fully encompass rock mass behaviour. Hence, field and laboratory testing must be conducted before implementing large geotechnical, geothermal, or infrastructure projects. Figure 9 depicts all permeability results from the field and laboratory in a probability density plot. There is an apparent discrepancy between the two scales. The laboratory-based results show that facies one, two, and three are similar, ranging from  $10^{-19}$  m² to high  $10^{-17}$  m² in permeability. Only facies four reaches a broader range from  $10^{-16}$  m² to  $10^{-13}$  m². The field data suggest an overall wider spread of permeabilities. Here, the values vary significantly, from  $10^{-17}$  m² to nearly  $10^{-10}$  m². Facies three and four from field testing show a right-skewed distribution, but are generally unimodal, while laboratory data also displays a bimodal distribution. As a few tens of cubic centimeters of a laboratory plug cannot include significant apertures, open fractures, or all relevant structures larger than the specimen itself, these features remain mostly untested on the lab scale.


The trend of field permeability being higher than laboratory investigations, and that permeability generally rises with increasing scale (up to a certain point), was documented by authors before (Clauser, 1992; Guéguen et al., 1996; Guimerà and Carrera, 2000; Illman, 2006; Martinez-Landa and Carrera, 2005; Münger, 2020; Neuman and Di Federico, 2003). This discrepancy is even more pronounced in fault zones, as heterogeneity and anisotropy vary strongly over short distances. Initially, it could be assumed that this observation is purely due to sampling bias (Hunt, 2003); however, research has shown that in fractured media, the distinction between sampling artifacts and the intrinsic scale dependence of hydraulic properties is not clear-cut. At small scales, apparent variability largely reflects incomplete sampling of the fracture network, while at

Figure 9: Probability density plot of permeability for different facies on field and laboratory scale. N indicates the number of data points per facies

larger scales, the connected structure of fractures and the distribution of transmissivities exert a genuine control on effective permeability. Thus, what is often termed 'sampling bias' is in practice the mechanism through which the intrinsic network properties manifest as scale-dependent hydraulic behavior (Davy et al., 2023). Figure 10 presents a collection of Bedretto-related data from various authors (David et al., 2020; Lützenkirchen and Loew, 2011; Ma et al., 2022; Masset and Loew, 2010; Münger, 2020; Ofterdinger, 2001; Osten et al., 2024; Volpe et al., 2023), along with data from this study. The known trend of varying permeability with scale can be directly observed in this accumulation of data. Laboratory campaigns, even when testing specimens that originate directly from the vicinity of a fault core, rarely exceed  $10^{-16}$  m², except for one highly porous sample from this study. Permeability values are significantly higher when working with field data. Varying interval lengths, connected structures, or fractures with wide-open apertures around the fault core can exhibit high permeabilities reaching  $10^{-10}$  m². Also, the data spread of values generated during field testing is more significant when ignoring the outlier from facies four. Figure 10 shows that after permeability reaches its maximum during in-situ borehole testing, values appear to decrease again with

rising scale. A schematic was added to the figure, depicting a conceptual scheme that characterizes faults into different types of fluid flow behaviour (Caine et al., 1996).

Figure 10: Permeability varying with scale. Bars mark the minimum and maximum ranges when several data points are available. Triangles indicate single values. All cited authors worked with data from the Bedretto tunnel.

## 5.2 MC fault zone

Figure 11 depicts the hydraulic architecture of the fault zone. The crosshole testing indicated that the major fluid flow directly passes through the fault core, serving as the main flow path. In BO6, fracture sets appear to be more clustered compared to BO5. As the fault zone displays a weakly developed damage zone, it can be categorized as a mix between distributed and localized conduit after Caine's (1996) definition. The observations from logging and hydraulic testing suggest that the thickness of the various facies likely varies both down-dip and along strike, and that the fault core is not sufficiently developed to act as an effective barrier to fluid flow. Flow dimensions in Fig. 11 indicate values between 1.95 and 1.58, mostly representing nearly radial flow. The fault core from BO6 displays a mixed-type flow dimension, which we call pseudo-channelized flow, as it neither represents pure radial nor fully channelized. This observation can be linked to the larger amount of clustering in BO6. It becomes evident that flow dimensions vary strongly over short distances, and the high heterogeneity makes it challenging to use this parameter for precise flow architecture predictions. As the facies appear to have branched off features, the fault core can be directly bordered by less disturbed facies, like zero or one. In that case, the core would not be surrounded by a real damage zone and could be categorized as a localized conduit (BO6). When a higher disturbance facies encloses the core, it

Figure 11: Illustration of major flow paths and structural as well as hydraulic boundaries. Flow dimensions are indicated along the boreholes. Due to the varying plunges of the wellbores, meters along boreholes are given from a plan view reference, and values in brackets indicate real logging depth.

acts as a damage zone (even though poorly developed), allowing the classification as a distributed conduit (BO5). Figure 12 depicts a conceptual three-dimensional sketch of how the investigated rock volume looks. It illustrates some highly conductive branches while at the same showing no real damage zone. The MC fault can be described as an immature fault zone developed on a pre-existing high fracture intensity zone. As fluid flow in crystalline lithologies is primarily dictated by open or partially open fractures, the high-intensity fracture segments with visible aperture are the driving factor in the fault zone fluid flow. Fracture sets can propagate across facies boundaries, enhancing the permeability of facies that would otherwise appear undisturbed. The permeability of the connected structures yielded a low 10<sup>-15</sup> m<sup>2</sup>, placing it on the lower end of field-scale


determined values. This can be explained by some of the pumping fluid dissipating into the weakly developed damage zone, or the fault core between the two wells might host some localized structures, like gouge or compacted cataclastic zones, which affect the flow path. Yet, the core behaves as a transmissive conduit that sustains elevated pore pressures, indicative of confined conditions. In contrast, the poorly connected, low-permeability damage zone remains closer to hydrostatic, showing lower pressures. The resulting pattern reflects hydraulic compartmentalization: a pressurized core bounded by relatively isolated flanks.

Figure 12: Sketch of the MC fault zone with boreholes five and six intersecting. Facies are highlighted in their respective colors. Lenses or local patches of gouge can be found close to the fault core. Note that the figure is not to scale and should only facilitate three-dimensional depiction.

https://doi.org/10.5194/egusphere-2025-4733 Preprint. Discussion started: 20 October 2025

© Author(s) 2025. CC BY 4.0 License.




# 6 Conclusion and outlook

This study underscores the fundamental role of scale in determining the hydraulic properties of crystalline fault zones. Laboratory tests consistently yielded low permeability values, while field investigations revealed a much wider spread, extending up to  $10^{-10}$  m<sup>2</sup>. This scale effect is well documented in fractured rock systems and arises partly from sampling limitations of small specimens, which cannot capture large apertures or connected fracture networks. However, it also reflects intrinsic network properties: as the scale of investigation increases, the connectivity of transmissive structures exerts genuine control on effective permeability. The growing spread of field-derived values therefore illustrates that permeability is not a fixed property, but a scale-dependent parameter shaped by both sampling bias and the intrinsic heterogeneity of fault structures.

Crosshole testing in the MC fault indicates that the main flow pathway is concentrated in the fault core, with the surrounding damage zone only being weakly developed. Local differences between boreholes show that fracture clustering can generate distinct flow regimes, ranging from nearly radial to pseudo-channelized. Such variability over short distances complicates the use of flow dimensions as predictive indicators and reinforces the need for direct testing at multiple scales.

From a practical perspective, the lateral extent of the fault (>100 m; Achtziger-Zupančič et al. (2024)) requires that further test volumes on the opposite side of the Bedretto tunnel should be treated as hydraulically independent. Hydraulic testing is essential for selecting appropriate packed-off intervals in stimulation campaigns, since candidate zones must both sustain pressure build-up and provide sufficient transmissivity. Injection pressures, flow rates, and pumping durations can only be defined reliably on the basis of such site-specific characterization, highlighting that careful pre-stimulation testing is an indispensable step in fault-zone investigations. For controlled stimulation and earthquake triggering, this means that the chosen injection intervals should have few to no neighbouring fractures outside the packed-off interval, to restrict fluid flow to any parts other than the fault core. Additionally, several boreholes allow for cross-hole testing and simultaneous injection, creating several pressure front 'patches' which could hydraulically connect to a larger diffusion-dominated region, likely increasing the chance of inducing a tremor.

Finally, while pathways other than the fault core are difficult to isolate directly, discrete fracture network (DFN) modelling offers a powerful means of bridging structural observations and hydraulic data. This technique accounts for fracture connectivity and variability across scales. DFN approaches can help translate localized measurements into a broader understanding of fault-zone architecture and its role in controlling subsurface fluid flow, and provide interesting applications for further research.


# Appendix A

Figure A1: Comparison of WIP porosity measured on entire core segments compared to plugs. Horizontal and vertical bars indicate standard deviation. The 1:1 comparison line is marked as a dashed line.

Figure A2: (a) Comparison of Vp measured on core segments and plugs. (b) Comparison between WIP and He-pycnometry performed on plugs. (c) Comparison between core Vp velocity and core porosity measured with WIP. (d) Comparison between plug Vp velocity and plug porosity measured with WIP. The 1:1 comparison line is marked as a dashed line. The data points are facies color-coded.

# A.1 XRD and CT scans

The x-ray diffraction allowed the plotting of a QAP diagram, which illustrates how the different facies along the borehole vary from a mineralogical viewpoint. 20 wt% of internal standard was added to the bulk materials as an accuracy control. The CT scans served as supplementary material, highlighting the different appearances of the pore space, as well as providing additional porosity estimates.

515 Figure A3: XRD data plotted on a ternary QAP plot for intrusive igneous rocks. Samples originate from borehole five and highlight the mineralogical composition of specimens across a crystalline fault zone.

| Depth (m)   | Voxel resolution (µm) | Porosity (%) |
|-------------|-----------------------|--------------|
| 37.53-37.57 | 15                    | 0.26         |
| 38.58-38.62 | 15                    | 0.66         |
| 41.31-41.35 | 16.87                 | 0.72         |
| 43.35-43.38 | 22.50                 | 7.97         |
| 43.40-43.44 | 15                    | 4.47         |

520 Figure A4: CT scan results with visualized pore volume of four samples. The table indicates voxel resolution and computed porosities.

Code availability

Python codes can be provided upon email request to the main author. No other publicly non-available software was used.

Data availability

Datasets can be found under DOI: https://doi.org/10.5281/zenodo.17233183

Author contribution

Conceptualization: TS, MJ; data curation: TS, AC, AZ, GP; formal analysis: TS, AC, AZ, GP, TSM, HC; funding acquisition: DG, SW, MC, FA; investigation: TS, AC, TSM, HC, YG; methodology: TS, TSM; project administration: MJ, VG, MH, M.A. M; resources: MH, TSM, HC; software: TS; supervision: MJ; validation: TS, MJ, TSM; visualization: TS, AC, HC; writing - original draft: TS; writing - review and editing: TS, MJ, AC, VG, M.A. M, TSM, YG, FA.

Competing interests

The authors declare that they have no conflict of interest.

Financial support

This project has received funding from the European Research Council (ERC) project FEAR (grant 856559) under the European Union's Horizon 2020 research and innovation programme.

545

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
