# Peer review of "Multi-Scale Hydraulic and Petrophysical Characterization of a Heterogeneous Fault Zone in the Gotthard Massif's Crystalline Basement"

_EGUsphere, 2025_

## Author Comment (AC3)

**Report of the replies to the reviews and further unrelated changes**

P 1-3: Reply to the review of Giacomo Medici CC1
P 3-6: Reply to the review of the Anonymous Referee #1 RC1
P 6-8: Reply to the review of the Anonymous Referee #2 RC2

Dear Giacomo Medici,

We want to thank you very much for the thorough review and the critical comments. Your constructive suggestions and corrections improve the quality of our manuscript. On the following pages, we present our changes and corrections to your individual comments.

Kind regards, Tom Schaber and co-authors

A) Line 49. "to field projects reaching dimensions in the decametre range". Please, integrate recent literature that combines laboratory scale measurements to hydraulic tests with decametre ranges including straddle packer systems:

- Quinn, P., Cherry, J. A., Parker, B.L. 2012. Hydraulic testing using a versatile straddle packer system for improved transmissivity estimation in fractured-rock boreholes. Hydrogeology Journal, 20(8), 1529-1547.

- Agbotui, P.Y., Firouzbehi, F., Medici, G. 2025. Review of effective porosity in sandstone aquifers: insights for representation of contaminant transport.  Sustainability, 17, (14), 6469.

→ The cited references by the author are supposed to highlight the work that was already done in this specific field, mostly related to fault zone hydraulics, fault plane reactivation, and seismicity. We think the 2012 publication would be best placed in the methodology part, serving as an add-on to the hydraulic field testing. The suggested 2025 publication focuses on effective porosity and contaminant transport in fractured sandstone aquifers and does not specifically address decameter-scale straddle-packer testing or fault-zone hydraulic stimulation in crystalline rock; we therefore consider it less directly applicable to the context of line 49. However, it could make sense to include it in the general scale-effect discussion as a recent multiscale hydrogeology review."

B) Line 68. Clearly state the aim of your research.

Line 68. Specify and describe the objectives of your research by using numbers (e.g., i, ii, and iii).

→ To make this point clearer, we are going to add a small piece of text to the beginning of the same paragraph, explicitly highlighting that the hydraulic and partly structural characterization of the fault zone is the exact aim of this work.

C) Line 111. "mini-frac campaign". Specify the techniques used during this campaign.

→ A short explanation about hydraulic fracturing and HTPF testing will be added.

D) Line 199. Have you always used the Darcy's Law in the manuscript for permeability? Or another one before?

→ The paragraph you highlighted deals with the laboratory testing, which incorporated both classic steady-state flow measurements and pulse-decay experiments after Brace (1968). Both methods are built on Darcy's law.

E) Line 226. Optical and acoustic televiewer logs. Can you specify the logging speed? It strongly affects the quality of the images and the structure picking.

→ We agree on the importance of proper logging procedures and documentation. Logging speeds will be added to that very line.

F) Line 277. What do you mean by "channelized flow" opposed to linear? Non-Darcian / non laminar? These words are also used in karst hydrology for conduit flow in cavities with approximate pipe-shape.

→ Going after the Generalized Radial Flow Model after Barker (1988), what we call "channelized flow" is essentially something similar to linear flow (pipe-shaped as you stated). Using the term channelized describes a somewhat linear flow regime behaviour that follows preferential pathways. It can be envisioned like "where the flow goes". This channeling can occur due to fracture clustering and overlapping or intersecting fractures, creating small, voluminous pathways through which the flow passes. So yes, it sounds similar to what you stated from karst hydrology.

G) Figure 4a. Make the letters much larger.

→ Yes, will be implemented.

H) Figure 10. Important figure that can attract attention to a public of either geo-physicist or hydro-geologists. Please, try to improve. Who knows it might be incorporated in textbooks!

Figure 10. Make the words larger.

Figure 10. "Laboratory", "in situ-borehole", and "meso-macro scale". Please, specify the techniques used.

→ The figure was adequately improved, and an additional line in the figure caption will refer to the different techniques used.

I) Figure 11. I remind explanation for channelized flow.

→ Explanation see point F). The authors are going to make some small visual changes to the figure, but no changes regarding the content.
* * *
Dear Anonymous Referee #1 RC1,

We want to thank you very much for the thorough review and the critical comments. Your constructive suggestions and corrections greatly improve the quality of our manuscript. On the following pages, we present our changes and corrections to your individual comments.

Kind regards, Tom Schaber and co-authors

A) Composition of the material was discussed, and from the descriptions of the different facies that the mineral composition changes with proximity to the fault core. Since permeability is strongly controlled by mineralogy, I think the composition of the samples needs to be reported. I see in the appendix that there are XRD measurements, so please present the modal abundances of each mineral (abundances of < 5% don't need to be reported).

→ A table was added to the appendix highlighting the requested abundances.

B) The text on L376-379 reports that the effect of increasing effective stress during the lab permeability tests is larger for the more impermeable samples compared to the more permeable ones. This is a counterintuitive interpretation that I don't think is correct, and is probably and effect of different scales in Fig. 8. The decrease in permeability with increasing effective stress in Fig. 8a and 8b look large on the log scale, but they will be negligible compared to the samples in 8d that have an absolute permeability value on the order of $10^{-12}$ or even $10^{-16}$ m$^2$. Conversely, what looks like a negligibly small difference in 8d could still be orders of magnitude larger than any change in the other three panels. Because the absolute values of permeability are very small in low permeability samples, the figure actually shows that the effect of increasing stress is correspondingly small. I would refrain from comparing the stress sensitivity of the samples and only comment on the stress sensitivity for an individual facies or group of samples where the absolute values of permeability are on the same order.

→ Thanks a lot for this specific feedback! This highlights the need for absolutely clear and unambiguous communication, which was not the case in this paragraph. The reader might have picked up the impression that the authors are implying a general rule, which states that tighter facies generally appear to be more stress sensitive than already permeable ones. This is NOT the case, and was badly formulated. When looking at the relative drops of permeability across the effective stress stages, it becomes evident that the reduction in permeability is rather similar across facies 1, 2, and 3. The exception is the highest permeable sample in the entire data set ($10^{-13}$ m$^2$) in facies 4. That sample is characterized by vuggy, voluminous cavities of many cubic millimetres, which were clearly macroscopically visible. These voids were too large to be noticeably affected by any confining pressure, no matter the height of the effective stress. Hence, this sample's permeability does not change at all across the rising effective stress stages. All other samples either had much smaller cavities or only very small aperture fractures, or even microfractures, which the confining stress could easily act on and close. We rephrased these few sentences to make it unmistakably clear.

C) I couldn't quite follow the discussion on L404-424. The trend in Fig. 10 can be clearly attributed to sampling bias, so I am not sure what the argument is here. Is the argument that there is still a scale-dependence of permeability even without sampling bias? I can imagine that small-scale measurements of permeability will have a very wide range if the sampling bias were removed such that large fractures were also being sampled, but if the sampling density was large enough to approximate the fracture network distribution at large scales, the large-scale permeability could be accurately determined. Of course, this is difficult and not practical at the moment.

→ This paragraph tries to highlight the possibility that both a sampling bias and an actual change in permeability depending on scale can be present. This was investigated by Davy et al. (2024). They first summarized previous work on sampling bias and permeability scale dependencies, before presenting their own results. They claim that the scale-dependent permeability behaviour kicks in after the observed system reaches a certain percolation or connectivity transition. In their study, they worked with data from the Forsmark site (also crystalline bedrock) and found a transitional regime for the percolation at a scale of roughly 30 m. We believe that such a kind of scale dependency might also hold for our case. We added a more detailed explanation of this concept and made our writing more understandable so that it becomes more accessible to the reader.

D) L35: "types" rather than "appearances"

→ Yes, agreed.

E) L87-90: I'd like some more detail on how these RG1 and RG2 designations came about. Are they simply end members of the granite composition? Or are they from specific sampling sites or sampling strategies? Also, I would consider the proportion of the mineral types to be part of the composition, so it is confusing/misleading to say that the composition does not vary.

→ The terms RG1 and RG2 were introduced in detail by Rast et al. (2022). As this goes into greater depth and beyond the scope of this work, we decided not to place further emphasis on the mineral composition and keep the explanation broader. As we are purely dealing with RG1, we added a brief clarification explaining it in more detail.

F) L101-106: I think it makes sense to switch Fig. 2 and 3, and indicate where the borehole images came from.

→ Agreed, we switched the figures.

G) L127-128: What do you mean by cohesive or non-cohesive fractures? Cohesiveness is a mechanical behavior and not really something that can be described visually. Or do you mean open or filled fractures, or something similar?

→ Yes, cohesive in this case means sealed or cemented. For F0 and F1, mostly with phyllosilicates. This nomenclature was introduced by Achtziger-Zupancic et al. (2024) who originally defined the different facies types.

H) L173: What is the composition of the core samples? RG1 or RG2? Is there some basic XRD measurement available that quantifies the main mineral phases?

→ As stated in E), the volume of investigation is entirely situated in RG1, the equigranular type. For further and more detailed information about the mineral phases and XRD, we provide a table in the appendix as in A).

I) L181-183: I suppose this means that the core samples here are isotropic and do not need to be oriented? Or were they recovered at a specific orientation? I suggest adding a sample photo to Fig. 4.

→ The drill cores were unfortunately retrieved without orientation. Hence, it was not possible to test the parameters with respect to their true in-situ orientation, e.g., the stress field. 4 pictures of core pieces from facies 1-4 can be found in Figure 8.

J) L190: low permeability

→ Yes, agreed.

K) L311: due to…?

→ …due to the vuggy pores of facies 4.

L) L313: Please define $P_{10}$ and how it was determined.

→Yes, this was implemented in the same sentence.

M) L320,354: Please provide a facies legend for Fig.6 and 7

→Yes, this was added.

N) L451-452: This mention of pore pressures comes a bit out of nowhere, it would be helpful to re-state the values here and add some discussion to give the values context. What are the expected hydrostatic pore pressures? How large are the overpressures (i.e., what are the lambda values)?

→ Further clarification would make sense here, agreed. As the vertical stress acting around boreholes 5 and 6 is most likely not varying too much over these short lateral distances between the packed-off intervals (only a few tens of meters), it would be tricky to try and imply solid values for lambda, as overburden is not predictable to sufficient precision. That's why we are going to use the borehole mouth/tunnel floor as the reference point for the hydraulic head. This also avoids the problem of a lack of knowledge about the regional water table. This was explained in a short new paragraph.
* * *
Dear Anonymous Referee #2 RC2,

We want to thank you very much for the thorough review and the critical comments. Your constructive suggestions and corrections greatly improve the quality of our manuscript. On the following pages, we present our changes and corrections to your individual comments.

Kind regards, Tom Schaber and co-authors

A) Line 35-59: The opening paragraph starts with a very broad overview of earthquake hazards and societal impacts, then illustrate the difficulty in predicting earthquakes and the reasons behind. However, the transition from earthquake to hydraulic testing feels somewhat abrupt, there is a missing logical link explaining why hydraulic characterization is a critical step for the stimulation experiments. Also, the current content lists permeability (line 45) as one of many factors in the complex system, it would be beneficial to emphasize it to have a smoother transition of the topics.

→The authors partly agree. Sentences as "To develop an appropriate injection scheme and protocol for the stimulation experiments, the fault zone must be hydraulically and structurally characterized beforehand." (line 60) or "This allowed a hydraulic characterization and provided

a collection of in-situ parameters such as permeability, pore pressure, or injectivity" (line 62) already shed some light on these aspects. We provided further clarification. The sentence in line 45 can be changed accordingly, in order to put the permeability more in focus.

B) Line 309-311: "Moving away from the … due.' This is an unfinished sentence, please recheck it.

→…due to the vuggy pores of facies 4.

C) Figures 6 and 7: These figures would be much more readable if the facies legend were included directly, avoiding the need to scroll back to previous sections.

→ Yes, was added.

D) Line 376-379: The text states that effective stress impacts tighter facies more intensely than high-permeability ones. However, this conclusion is difficult to verify visually because the plots in Figure 8 span vastly different permeability ranges. A slight visual reduction in the high-permeability samples (Fig. 8d) might actually correspond to a much larger absolute reduction compared to the tighter facies. I suggest clarifying this conclusion to avoid potential misinterpretation.

→This was also mentioned by the anonymous reviewer No. 1. Thanks a lot for this specific feedback! This highlights the need for absolutely clear and unambiguous communication, which was not the case in this paragraph. The reader might have picked up the impression that the authors are implying a general rule, which states that tighter facies generally appear to be more stress sensitive than already permeable ones. This is NOT the case, and was badly formulated. When looking at the relative drops of permeability across the effective stress stages, it becomes evident that the reduction in permeability is rather similar across facies 1, 2, and 3. The exception is the highest permeable sample in the entire data set ($10^{-13}$ m$^2$) in facies 4. That sample is characterized by vuggy, voluminous cavities of many cubic millimetres, which were clearly macroscopically visible. These voids were too large to be noticeably affected by any confining pressure, no matter the height of the effective stress. Hence, this sample's permeability does not change at all across the rising effective stress stages. All other samples either had much smaller cavities or only very small aperture fractures, or even microfractures, which the confining stress could easily act on and close. We rephrased these few sentences to make it unmistakably clear.